REGISTERED REPORT PROTOCOL

# The social learning and development of intra- and inter-ethnic sharing norms in the Congo Basin: A registered report protocol

Sarah Pope-Caldwell[1]☸*, Sheina Lew-Levy[2,3]☸*, Luke Maurits[1], Adam H. Boyette[3], Kate Ellis-Davies[4], Daniel Haun[1,5], Harriet Over[6], Bailey R. House[6]

1 Department of Comparative Cultural Psychology, Max Planck Institute for Evolutionary Anthropology, Leipzig, Germany, 2 Department of Psychology, Durham University, Durham, United Kingdom, 3 Department of Human Behavior, Ecology, and Culture, Max Planck Institute for Evolutionary Anthropology, Leipzig, Germany, 4 Department of Psychology, Swansea University, Swansea, United Kingdom, 5 Leipzig Research Centre for Early Child Development, Leipzig University, Leipzig, Germany, 6 Department of Psychology, University of York, York, United Kingdom

☸ These authors contributed equally to this work.
* sarah_pope@eva.mpg.de (SPC); sheina.lew-levy@durham.ac.uk (SLL)

## Abstract

Compared to other species, the extent of human cooperation is unparalleled. Such cooperation is coordinated between community members via social norms. Developmental research has demonstrated that very young children are sensitive to social norms, and that social norms are internalized by middle childhood. Most research on social norm acquisition has focused on norms that modulated intra-group cooperation. Yet around the world, multi-ethnic communities also cooperate, and this cooperation is often shaped by distinct inter-group social norms. In the present study, we will investigate whether inter-ethnic and intra-ethnic social norm acquisition follows the same, or distinct, developmental trajectories. Specifically, we will work with BaYaka foragers and Bandongo fisher-farmers who inhabit multi-ethnic villages in the Republic of the Congo. In these villages, inter-ethnic cooperation is regulated by sharing norms. Through interviews with adult participants, we will provide the first descriptive account of the timing and mechanism by which BaYaka and Bandongo learn to share with out-group members. Children (5–17 years) and adults (17+ years) will also participate in a modified Dictator Game to investigate the developmental trajectories of children's intra- and inter-ethnic sharing choices. Based on our ethnographic knowledge of the participating communities, we predict that children's intra-ethnic sharing choices in the Dictator Game will match those of adults at an earlier age than their inter-ethnic sharing choices. We will analyze our data using logistic Bayesian modelling.

## Introduction

Cooperation is central to the success of our species [1, 2]. Humans cooperate across a range of daily activities including subsistence [3], food sharing [4, 5], childcare [6, 7], and knowledge transmission [8, 9]. Compared to other primates, human cooperation is unique in the extent

**Data Availability Statement:** All data analysis scripts are available in our GitHub repository at https://github.com/ccp-eva/sharing-norms. There is no data associated with this registered report protocol. Data collected as part of this study will be published alongside the completed registered report.

**Funding:** SLL was funded by a postdoctoral fellowship from the Alexander von Humboldt Foundation. DH was funded by the Max Planck Society for the Advancement of Science. The funders had no role in study design, data collection and analysis, decision to publish, or preparation of the manuscript.

**Competing interests:** The authors have declared that no competing interests exist.

to which it varies across communities [10, 11], and in that we cooperate with many unrelated and even unknown individuals [12, 13]. Social norms, or "mutual agreements or commitments about the way that individuals ought to behave in certain situations" [8], ensure that cooperation is coordinated between community members [14–16]. In turn, ethnic markers such as language, behaviours, and styles of dress help community members identify each other, hence facilitating in-group cooperation [14, 17].

Most research into the development of social norms has focused on intra-group cooperation. These studies have demonstrated that children are sensitive to social norms by the age of three [18], that children internalize social norms by middle childhood [19], and that children as young as three preferentially cooperate with in-group members [20]. Yet, around the world, many individuals live in multi-ethnic communities characterized by inter-group cooperation [21–24]. Inter-group cooperation may help communities manage risks associated with resource shortfalls and provide access to nonlocally available resources [25–27]. In such communities, individuals not only have social norms for cooperating with individuals *within* their ethnic groups, but *between* them.

In the present paper, we aim to investigate the development of intra- and inter-ethnic social norms in multi-ethnic villages inhabited by BaYaka foragers and Bandongo fisher-farmers in the Republic of the Congo. These communities regularly engage in inter-ethnic cooperation regulated by sharing norms. In this study, we will provide the first descriptive account of the timing and mechanisms by which BaYaka and Bandongo learn to share with out-group members via interviews with adult participants. We will also use a modified Dictator Game to investigate the developmental trajectories of children's intra- and inter-ethnic sharing choices. Considerable research has been done to develop experimental tasks measuring sharing across a wide range of ages and cultures, providing a firm methodological footing for this experimental paradigm. In what follows, we summarize findings from previous studies on the development of intra-group social norms and inter-group biases. We then describe the context in which the present study will take place.

## Developing social norms

Prosocial behaviour emerges in infancy [28] and increases in both frequency and sophistication between early childhood and adolescence [29]. Children as young as three rapidly infer the presence of norms, protesting the incorrect usage of an object after having seen it used 'correctly' by an adult model only once [18]. By age five, children spontaneously generate their own norms in novel collaborative games [30], and protest norm violations that would benefit them in a competitive game [18]. German and American three- to five-year-old children show a clear willingness to conform to the behaviour of others [31–33].

Societal variation in prosocial behaviour begins to increase around 7–10 years of age. For example, studies have found increasing societal differences in both generosity in a binary Dictator Game [19, 29, 34] and advantageous inequity in the Inequity Aversion Game [35]. Children in middle childhood also modify their prosocial behaviour in response to normative information. For example, 6- to 9-year-old German children were more likely to share in a binary Dictator Game when their knowledge of local norms was primed (i.e. when they were told they could "share like they think they ought to share") relative to when their own preferences were primed (i.e. when they were told that they could "share as they wished") [36]. This suggests that these children held knowledge of a norm specifying that they "ought to share". Researchers have also found that 6- to 10-year-old German and British children's prosocial behaviour in the binary Dictator Game becomes increasingly influenced by norms as they age [37, 38]. Taken together, these findings suggest that by middle childhood, children become

increasingly sensitive to, and likely to conform to, community-specific norms. However, how social learning contributes to variation in the timing of norm acquisition has been infrequently explored.

## Developing in-group preferences

Inter-group bias appears early in development. American five-year-olds show consistent preferences for members of their own age group [39], gender [40, 41] and language group [42]. Sensitivity to group membership cues also shape children's social learning. Buttelmann and colleagues [43] showed that 14-month-old German children were more likely to imitate the actions of speakers of their own language than speakers of a different language. In comparison, when presented with an action performed by out-group members, 5-year-olds were more likely to perform a contrasting action than the one observed [44].

Children are more likely to act cooperatively with in-group members [42, 45–47]. For example, 2.5 year-old American children are more likely to share toys with native language speakers [20]. Children from western cultural contexts are also more likely to help and share with members of their own ethnic groups [46, 48–50]. Even arbitrary or transient in-groups result in increased prosocial behaviour. American preschoolers preferentially allocate resources to randomly classified in-group members marked by armband and sticker colors [51]. Yet, norms may play an important role in regulating inter-group sharing. When equitable, Swiss second-graders adhere to suggested in-group and out-group sharing norms [52]. Inter-group cooperation may be enhanced in communities where strong social norms regulate inter-ethnic interactions [53].

## Learning about social norms and group membership

Ethnographic research suggest that several social learning mechanisms contribute to children's growing awareness of community-specific social norms, including sharing norms [54, 55]. Parents play an active role in teaching children to share in early life. For example, when Kalahari San eight-month-olds give objects to others, parents actively encourage them [56]. In early childhood, Indian Nayaka parents send children to distribute plates of food to other households [57]. When children refuse to share, Central African Aka caregivers may withhold food, gossip about them, hit them, or insult them [58]. In some Chinese schools, teachers actively provide instruction related to fairness through disciplinary and motivational interventions, peer comparisons, and moral comments [59]. Children also learn sharing norms in child-only groups. For example, a Congolese BaYaka child may carefully dole out tiny portions of food during play, sending these portions to other children in a manner emulating adult sharing [60]. In peer groups, Israeli children participate in ritualized sharing of candy and other treats [61]. While foraging, Tanzanian Hadza children share food with their peers, and abstain from consuming food so that they can share them with their caregivers upon return to camp [62, 63].

Social learning also plays an important role in children's acquisition of beliefs about in- and out-group membership and their attitudes towards them [64–67]. In terms of the acquisition of stereotypical beliefs, a large body of research has shown that children are exposed to cultural stereotypes in conversation with parents as well as through the broader culture [67]. In terms of the acquisition of inter-group attitudes, a comprehensive meta-analysis of more than 45,000 parent-child dyads from predominantly western cultural contexts reported a moderate and positive relationship between the attitudes of children and those of their parents [68]. Supporting this view, Skinner and colleagues [69] demonstrated that observing negative non-verbal behaviour towards a stranger is sufficient to lead American children to hold a negative attitude

towards that person and towards that person's friends. Interactions between children may also shape their out-group attitudes; Peruvian Matsigenka children adopted the norms of their Mestizo neighbours through sustained inter-ethnic interactions, primarily at school [70]. However, we know of no ethnographic studies that have described how children learn inter-ethnic social norms, despite their importance to inter-group cooperation.

## Ethnographic setting

Our study will explore the development of intra- and inter-ethnic sharing norms among BaYaka and Bandongo inhabiting two villages alongside the Motaba river in the Likouala Department of the Republic of the Congo [71]. Bandongo are primarily fisher-farmers who also participate in hunting and trapping [72]. BaYaka are foragers who primarily collect honey, wild yams, mushroom, fish, wild game, and other forest products, supplemented by cultigens from low-intensity gardens [73, 74]. BaYaka and Bandongo primarily use linguistic (Yaka and Bondongo/Lingala) and behavioural ethnic markers to distinguish between their communities. BaYaka view Bandongo as accumulators of wealth, hierarchical, and as claimants of forest areas as their own [75]. Bandongo identify BaYaka based on their sociability, lack of food reserves (reflecting their immediate-return economy), and their knowledge of the forest [75].

BaYaka intra-group sharing norms are organized around generalized reciprocity. Consistent with their strong egalitarian ethos, having a resource is understood by BaYaka as having an obligation to share it, and an expectation that others have the right to demand it [76]. Food sharing norms are formalized into specific food taboos about how hunters allocate their kills, based on seniority, gender, and specific roles during the hunt. Portions of cooked meals are typically shared according to kinship, residential proximity, reciprocity, and need [77]. Sharing of most resources is unconditional and failing to share is not only viewed as inviting social discord into the community, but also as angering the forest, thus threatening the future availability of resources [78].

Among the many ethnic groups of farming and fishing peoples in the region, including Bandongo, sharing norms reflect the cultural values of family communalism and status hierarchy [79]. Resource production and consumption typically occurs along the patriline. Sharing within households is governed by gender and age. Resources are considered the private property of the family. Within extended families, sharing is governed by norms of obligation to specific relatives (e.g., elders, in-laws). Between families, resources are exchanged via barter or sale, and sharing is largely constrained to community-wide events, such as funerals and rites of passage. Generally, people keep track of debts either formally or informally. Individuals or families deemed as having much more than others are accused of using witchcraft. Such threats help avoid disproportionate accumulation.

Sharing between BaYaka and Bandongo occurs in the context of economic exchange relationships, typically institutionalized through fictive kinship [80]. BaYaka men are often hired as shotgun hunters for Bandongo. In these contexts, BaYaka receive the hunter's portion of meat, including the head, the tail, and the guts. They also receive a pre-agreed gift, such as a headlamp or clothing [81]. BaYaka women routinely help build Bandongo houses, in exchange for palm wine and a manioc dish called *jabuka* (Yaka) or *pondu* (Lingala). BaYaka women collect caterpillars which they exchange for baby clothes and bassinets. Both BaYaka men and women contribute to farming labour in exchange for cultigens such as manioc, plantains, and corn. Finally, BaYaka and Bandongo inherit overlapping sections of the forest. Forest resources are jointly managed through harvesting and sharing rules. It is important to note that these sharing norms are not without contention. Conflicts can arise when one party

considers resources to be unfairly shared, when debts have not been paid, or when an area of the forest no longer produces sufficient resources. In such cases, institutions such as council meetings and *nganga* healing ceremonies help mitigate potential inter-ethnic conflict [82].

BaYaka children begin to learn intra-ethnic sharing norms in infancy, and socialization of sharing continues throughout childhood [83]. For example, children say their mothers taught them to share by showing them how to allocate portions of the evening meal [58]. Women then call upon children to distribute these plates to specific members of the community [83]. Sharing norms are also likely reinforced though the practice of demand sharing, which means that anyone has a right to ask another for a portion of a resource, and they are obligated to abide. Enforcement of sharing norms is indirect, and typically individual transgressions are not specifically sanctioned. Rather, adults will refer to improper sharing as the cause of failures to catch game on hunts, or they will denounce selfishness in a general way during public speeches (*mosambo*) or through satirical reenactments of selfish or improper behavior (*moadjo*) [82, 84]. While less is known about how Bandongo learn sharing norms, children, especially girls, are involved in food preparation from early in life and have the same opportunities BaYaka children do to observe their parents' sharing practices. As noted above, accusations of witchcraft are one serious form of public norm enforcement that children would witness.

How or when children learn inter-ethnic sharing norms is less understood. In early and middle childhood, BaYaka and Bandongo children may accompany their parents and observe while these participate in inter-ethnic labour or exchanges. However, it is not until early adolescence that inter-ethnic sharing becomes common. Bandongo adolescents hire BaYaka peers to go hunting to raise sufficient funds for the upcoming school year, often living in forest camps together for extended periods of time. BaYaka and Bandongo adolescent girls accompany their mothers to Bandongo fields. Bandongo pre-adolescent and adolescent children are also sometimes sent to collect debts from BaYaka for their parents. Such experiences provide extensive opportunities for adolescents to interact with out-group members, learn about each other's sharing norms, observe their parents barter and trade, and participate in exchanges themselves. Explicit teaching of inter-ethnic sharing norms may occur during *mosambo* and *moadjo* among BaYaka, when adults counsel adolescents in how to behave, and reprimand them when they have violated an inter-ethnic norm. Among the Bandongo, parents actively counsel their children regarding inter-ethnic sharing norms (Kandza, personal communication). Both BaYaka and Bandongo children also participate in village council meetings, where inter-ethnic norms are often discussed, and violations are resolved. The village crier (*mopandji sango*), who walks through both BaYaka and Bandongo neighbourhood in the evening sharing news of the day's activities as well as any decisions taken by the village council, often reminds community members to cooperate by respecting intra- and inter-ethnic sharing norms (Kandza, personal communication).

## The present study

As outlined, previous experimental research has demonstrated that children are sensitive to social norms and group membership in early childhood. By middle childhood, children have internalized community-specific social norms, leading to cross-cultural variation in behaviour. Social learning research further suggests that children develop social norms and inter-group attitudes from parents and other children via teaching, observation, and practice during play and work activities. Here, we add to this body of research by investigating the social learning and developmental trajectories of intra- and inter-ethnic sharing norms in a multi-ethnic community in the Republic of the Congo in which inter-ethnic cooperation is common.

Specifically, we aim to describe how, when, where, and from whom BaYaka and Bandongo learn inter-ethnic sharing norms via retrospective interviews with adults. We also aim to investigate how the development of adult-like intra- and inter-ethnic sharing norms are shaped by social learning opportunities using a modified Dictator Game. Our ethnographic work has demonstrated that BaYaka and Bandongo children learn intra-ethnic sharing norms in early childhood, whereas learning inter-ethnic sharing norms may occur more intensively in adolescence. We thus predict that children's intra-ethnic sharing choices in the Dictator Game will match those of adults at an earlier age than their inter-ethnic sharing choices.

## Methods

### Ethical approval

Ethical approval was obtained from Durham University Psychology ethics committee and from the Max Planck Group ethics committee. In-country permission will be obtained from the Institut de Recherche en Sciences Exactes et Naturelles and/or the Centre de Recherche en Economie et Sciences Humaines. Consent will be obtained in accordance with local cultural norms as established in previous field seasons. Specifically, community consent will be obtained during village meetings hosted by the Bandongo and BaYaka leaders. During these meetings, we will describe the goals of the study. We will emphasize that participation is not mandatory, and that individuals can withdraw from the study at any time without penalty. We will answer any questions the community may have. The community will decide by consensus whether we are permitted to conduct research in the villages.

Following community consent, we will seek individual verbal consent from adults. Consent scripts will be translated and back-translated into Lingala for Bandongo participants and Yaka for BaYaka participants. We will reiterate the goals of the research, the research procedure, the gifts received irrespective of participation, and that participants can withdraw at any time. We will answer any questions and ask for parent and/or guardian consent for child participation, where applicable.

Adult consent and child assent will be obtained again in the testing room immediately prior to administrating the experiment using a translated and back-translated assent script. In addition to verbal assent, we will attend to children's shyness or apparent discomfort. If, prior to the start of the experiment, or at any point throughout it, a child verbally or non-verbally signals that they do not wish to participate, we will stop the experiment and move on to the next participant.

Participants will receive culturally appropriate gifts commensurate with local sharing norms and the time they will spend working with us. Gifts may be mosquito nets, clothing, or schoolbooks, depending on community need and market availability. Upon completion of the research, we will make a poster of our findings which will be explained during a village meeting and given to each participating community.

### Participants

Data will be collected in two multi-ethnic villages of approximately 400 BaYaka and Bandongo inhabitants each with similar demographic profiles, subsistence strategies, access to markets, and distances from urban centers [85]. In our previous experimental and ethnographic work in these two villages, we have observed intra- and inter-ethnic sharing practices commensurate with regional social norms.

We will recruit 30 children (5–16 years) and 20 adults (17+ years) from each ethnic group in each village for a total of 200 participants. We aim for a sample evenly comprised of males and females within each ethnic group and village, and across age groups (grouped by 2 years

for children and 5 years for adults). To achieve this, we will develop a list of all consenting participants divided by age group, sex, and ethnicity. We will then randomly invite participants from the list to participate in the study. We will move down the list until 200 participants have (1) participated in all relevant elements of the study, and (2) have passed all comprehension checks embedded in the experiment (see below).

Our sample size is constrained to 200 individuals based on the population size of each village, and accounting for attrition attributed to the high levels of mobility in these communities, participant shyness which would preclude them from completing the experiment [85], and exclusion due to performance on the comprehension checks. Simulated datasets described in "Data Analysis" below suggest that this sample size is adequate to reliably estimate the order in which intra- and inter-ethnic sharing norms are acquired. Note that inter-ethnic cooperation generally, and sharing specifically, is highly regulated, easily observable in daily interactions, and widely practiced across a range of forest- and village-based contexts and resources. Thus, we do not *a priori* expect participant mobility and shyness to co-vary with inter-ethnic sharing norms, nor how these norms are acquired. Local perspectives on this potential limitation will nonetheless be elicited during the ethnographic interviews (see below).

## Inclusivity in global research

This project is only possible with the sustained support of local research assistants, who will conduct independent data collection as part of this project. Because of unknowns regarding staff availability at the time of data collection, it is currently not possible to definitively identify which research assistants will participate in the present study. Research assistants will be co-authors on the final registered report. Additional information regarding the ethical, cultural, and scientific considerations specific to inclusivity in global research is included in the S1 Checklist.

## Interview

An interview will be conducted with all participating BaYaka and Bandongo adults. Participants will be asked to describe how they learned sharing norms within and across ethnicity. These descriptions will be independently recoded into teaching and learning types adapted from Hewlett and Roulette [86] by the co-first authors (see Table 1 for definitions). After calculating reliability using Cohen's Kappa, we will resolve any disagreement by consensus. For both intra-ethnic and inter-ethnic sharing, we will also ask participants to free-list the cultural models (as categories, e.g. mother, friend) from whom they learned to share, and the contexts in which they learned to share [79]. We will also ask participants to identify the stage of childhood they began to learn to share. Life stages will follow local cultural understandings of child development which roughly map on to early childhood (*mwona/mwona moke*), middle

**Table 1. Description of teaching and learning categories used to code the interview responses, adapted from Hewlett and Roulette [86].**

| | |
|---|---|
| Demonstration | Caregiver shows the learner how to share. |
| Task assignment | Caregiver tasks the learner to share. |
| Instruction | Caregiver explains how sharing should or should not be undertaken, either *in situ* or through storytelling. |
| Observation/ Imitation | The learner observes the caregiver's sharing and/or imitates the caregiver's sharing. |
| Play | The learner emulates sharing during play. |

childhood (*mwona akoka*/*mwona ya mwamokolo*), and adolescence (*mopondi* (males), *ngondo* (girls)/*mwona ya mokolo*).

Following these structured questions, we will use an open-ended (i.e., ethnographic) format to understand secular changes in inter-ethnic sharing norms across generations, and whether mobility shapes inter-ethnic sharing norms. Questions will include: (1) How did the elders (*bakoko*) share across ethnicity? (2) How do you share across ethnicity? (3) Are children learning to share differently than in the past? (4) How did BaYaka and Bandongo interact in the past? (5) How have these interactions changed? (6) How do you resolve a conflict related to inter-ethnic sharing today? (7) How were these conflicts resolved in the past? (8) Do BaYaka/Bandongo who live in the forest full-time or almost full-time have different intra- and inter-ethnic sharing norms than those who spend more time in the village?

A short interview will be conducted with all participating BaYaka and Bandongo children, immediately after their participation in the Dictator Game (see below). Children will be asked to report whether they know how to share within and across ethnicity. Interview questions with a binary response (yes/no) which can be communicated non-verbally (e.g. by shaking one's head) can help overcome participant shyness common in experimental contexts [79]. In addition, we will ask child participants open-ended (i.e., ethnographic) questions including: (1) From whom did you learn intra-ethnic sharing norms? (2) From whom did you learn inter-ethnic sharing norms? (3) How do you share within ethnicity? (4) How do you share across ethnicity? While we expect fewer children will answer these questions due to shyness, a small sample of answers will nonetheless ensure that we are able to track secular trends regarding changes in the social learning of sharing norms over time.

## Dictator Game

All participants will play a binary choice Dictator Game, in which they make a series of choices between two predetermined payoff distributions. Rewards will be beads, which are used by both communities for hair adornment and as jewelry, and which are prized by males and females of all ages. To avoid bead colour affecting participation choice, all beads will be green. During the experiment, participants will choose to equally distribute two beads between themselves and another person (SHARE) or keep both beads for themselves (KEEP). This will be done over the course of two trials corresponding to two conditions in which the potential recipient is described as either the same ethnicity or a different ethnicity.

The Dictator Game has been successfully administered with neighbouring Aka forager children in the Central African Republic [19]. We have carefully designed aspects of this Dictator Game to be culturally sensitive and salient. In the classic Dictator Game, participants are presented with tokens that represent some unknown quantity. Participants exchange these tokens for rewards. To accommodate the immediate-return economy of BaYaka participants, we adjusted this Dictator Game to include the rewards—beads—as the currency in the game [34]. Further, all comprehension checks have been designed to require minimal verbal communication to help overcome participant shyness. We have opted for questions that can be answered via pointing, or in the case of counting, by holding up one or two fingers. All participants will be given two chances to pass the comprehension tests. If the participant fails any comprehension test thrice, the experiment will immediately end, and the participant will be excluded from the study.

**Materials.** The apparatus consists of two laminated paper trays 3.5 x 8.5 inch each with a yellow and purple circle on either end. Each tray represents a payoff distribution (SHARE or KEEP). Beads placed on the yellow circle facing participants are for participants, and beads placed on the purple circle further from participants are for recipient. Meeples, small

humanoid figurines, will be used to represent the participant and recipients, such that co-ethnic recipients will be the same color (blue or red) as the participant and vice versa. Participants and recipients will remain anonymous to each other. In each village, testing will occur in a quiet room in the research house. One BaYaka and one Bandongo research assistant will be trained to administer the experiment to co-ethnic participants. Experiment scripts have been translated and back-translated into Yaka for BaYaka participants, and Lingala for Bandongo participants. Research assistants will be guided through the experiment using the Open Data Kit application on a tablet [87]. Participant responses and choices will be recorded within the application. The location of the payoff distribution (left or right tray from the perspective of the participant), the meeple colour assigned to each ethnicity (blue or red), and the order of conditions (same ethnicity or different ethnicity recipient) will be randomized automatically within the application. We have opted to use these randomly-assigned visual markers of ethnicity because, as previously mentioned, BaYaka and Bandongo primarily distinguish each other through linguistic and behavioural ethnic markers, which do not straightforwardly translate to this experimental paradigm. Randomly-assigned colours have been used successfully as in- and out-group markers in previous experiments in the post-industrialized west [51], and are also used as part of team uniforms at the field site during inter-village football games. Thus, we expect that assigning meeple colours to ethnicity will be well understood in this context. Testing will be video recorded in case of equipment malfunction.

**Procedure.** Full procedural details are outlined in Fig 1. The testing procedure has 9 steps:

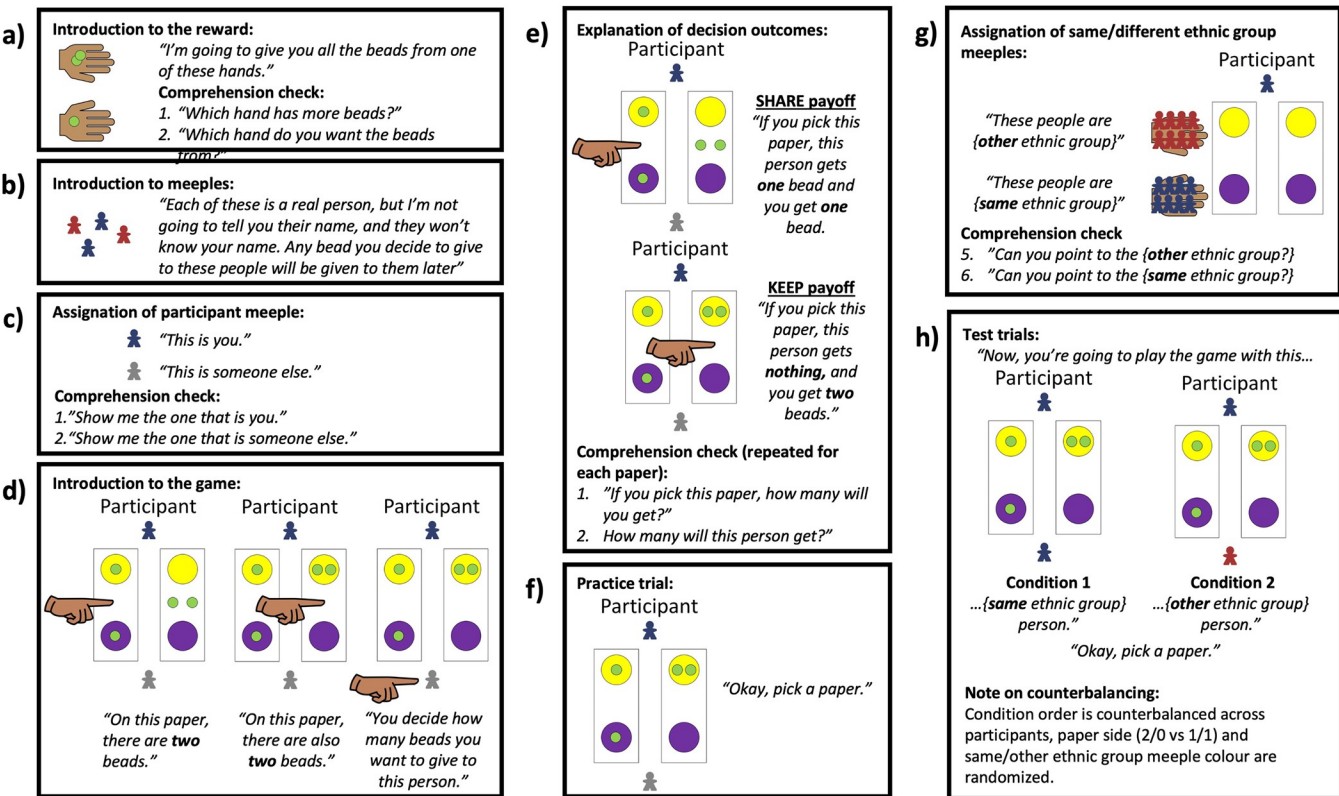

**Fig 1. Dictator Game procedural details.** a) introduction to reward; b) introduction to meeples and emphasis on their representation of real people; c) assignation of participant's meeple colour; d) introduction to game play; e) explanation of decision outcomes; f) practice trial; g) assignation of same/different ethnic group meeples; h) test trials.

1. <u>Introduction to the reward:</u> The experimenter hands the participant a cup. The experimenter places two beads in one hand, and one bead in the other. The experimenter holds out their hands to show the beads to participants. The experimenter says: "I'm going to give you all the beads from one of these hands."
   <u>Comprehension check:</u> The experimenter asks the participant to point to the hand with more beads. The experimenter then asks the participant to choose a hand from which to collect the beads. Participants will have **passed** the comprehension test if they correctly identify the hand with more beads.
   <u>Distribution of reward:</u> The experimenter places the participant's beads in their cup.

2. <u>Introduction to meeples:</u> The experimenter picks up a handful of meeples and shows them to the participant. The experimenter says: "Each of these is a real person, but I'm not going to tell you their name, and they won't know your name. Any bead you decide to give these people will be given to them later."

3. <u>Assignation of participant meeple:</u> The experimenter shows the participant a red OR blue meeple (counterbalanced) and a says: "This is you." The experimenter shows the participant a grey meeple and says: "This is someone else."
   <u>Comprehension check:</u> The experimenter asks the participant to point to the meeple that represents them. The experimenter then asks the participant to point to the meeple that represents someone else. Participants will have **passed** the comprehension test if they correctly identify the meeples.

4. <u>Introduction to the game:</u> The experimenter pulls out the trays, and places them such that the yellow circles face the participant. The experimenter places the participant's meeple in front of the yellow circle, and the grey meeple across from the participant's meeple, in front of the purple circles. The experimenter places two beads on the center of each tray. The experimenter points to the left OR right tray (counterbalanced) and says: "On this paper, there are two beads". The experimenter places one bead in the yellow circle, and one bead in the purple circle (SHARE payoff). The experimenter points to the second tray and says: "On this paper, there are also two beads." The experimenter places two beads in the yellow circle, and none in the purple circle (KEEP payoff). The experimenter points to the grey meeple and says: "You decide how many beads you want to give to this person."

5. <u>Explanation of decision outcomes:</u> The experimenter points to the SHARE tray and says: "If you pick this paper, this person gets one bead, and you get one bead." The experimenter then points to the KEEP tray and says: "If you pick this paper, this person gets nothing, and you get two beads."
   <u>Comprehension check:</u> The experimenter points to each of the SHARE and KEEP trays and asks the participant how many beads they will get, and how many beads the recipient will get, if they pick each tray. Participants will have **passed** if they correctly identify the number of beads they will keep and give for each tray.

6. <u>Practice trial:</u> The experimenter says: "Okay, pick a paper." The experimenter records the participant's response.
   <u>Distribution of reward:</u> The experimenter places the grey meeple in a cup, and places one or no beads (depending on the participant's choice) in that cup. The experimenter places the participant's beads in their cup.

7. <u>Assignation of same/different ethnic group meeples:</u> The experimenter picks up a handful of red meeples in one hand, and blue meeples in another. The experimenter shows the participant the red OR blue meeples (matching the participant's meeple colour) and says:

"These people are BaYaka OR Bandongo (matching the participant's ethnicity)." The experimenter shows the participant the other meeples and says: "These people are Bandongo OR BaYaka (contrasting the participant's meeple colour and ethnicity)."

Comprehension check: The experimenter asks the participant to point to the meeples of the participant's same ethnic group. The experimenter asks the participant to point to the meeples of the other ethnic group. Participants will have **passed** the comprehension test if they correctly identify the ethnicity of the meeples.

8. Test trials: The experimenter sets up the trays as in step 4. The experimenter then presents the two conditions, the order of which are counterbalanced across participants, with all participants participating in both conditions. The experimenter places a same-ethnicity meeple (condition 1) or an other-ethnicity meeple (condition 2) in front of the purple circles. The experimenter says: "Now, you're going to play the game with this BaYaka OR Bandongo person (same ethnicity for condition 1, other ethnicity for condition 2)." The experimenter says: "Okay, pick a paper." The experimenter records the participant's response. The experimenter places the recipient meeple in a cup, and places one or no beads (dependent on the participant's choice) in that cup. The experimenter places the participant's beads in their cup.

9. Interview questions & thank you: The participant is administered the interview questions and thanked for their participation.

## Data analysis

All analyses will be conducted in R [88].

**Qualitative interview analysis.** Answers to the open-ended (i.e., ethnographic) questions posed to adults and children regarding learning, cooperation, mobility, and inter-generation changes will be transcribed. Salient qualitative trends will be identified and summarized.

**Interview descriptive & exploratory statistics.** We will report the total number and percent of adult participants who reported learning intra- vs. inter-ethnic sharing via each teaching and learning type, separated by participant ethnicity. To explore how variation in teaching and learning types corresponded to acquiring intra- or inter-ethnic sharing norms, we will conduct 2X2 Fisher's exact tests on the BaYaka and Bandongo samples separately.

We will report the total number and percent of adult participants who reported learning intra- vs. inter-ethnic sharing via each category of cultural model (e.g., mother, friend), separated by participant ethnicity. To investigate variation in the cultural models who contribute to the learning of intra- vs. inter-ethnic sharing norms, we will conduct 2X2 Fisher's exact tests on the BaYaka and Bandongo samples separately.

We will report the total number and percent of adult participants who reported learning intra- vs. inter-ethnic sharing in each learning context, separated by participant ethnicity. To investigate variation in learning contexts for intra- vs. inter-ethnic sharing norms, we will conduct 2X2 Fisher's exact tests on the BaYaka and Bandongo samples separately.

We will report the total number and percent of adult participants who reported learning intra- vs. inter-ethnic sharing at each age stage, separated by participant ethnicity. To investigate variation in the age at which intra- vs. inter-ethnic sharing norms were acquired, we will conduct a 3X2 Fisher's exact test on the BaYaka and Bandongo samples separately.

Finally, we will report the total number and percent of child participants who reported knowing to share within and between ethnicities by age category (early childhood; 5–6 years, middle childhood; 7–12 years, adolescence, 13–17 years) separated by participant ethnicity. To investigate variation in knowledge of intra vs. inter-ethnic sharing norm acquisition by age, we will conduct a 3X2 Fisher's exact test on the BaYaka and Bandongo samples separately.

## Modelling experimental data

Experimental data will be analyzed using a Bayesian modelling approach [89], with estimation performed via Hamiltonian MCMC using Stan [90] and Rstan [91]. The developmental trajectories of sharing norms will be modelled using logistic curves, which transition smoothly between an initial infant and final adult probability for sharing. For each of the two societies, two curves representing the development of intra-ethnic sharing norms and two curves representing the development of inter-ethnic sharing norms will be fit to the data, totaling four curves. A single initial infant probability will be estimated across all four developmental trajectories, encoding the idea that the behaviour of infants, prior to socialization, is not expected to vary across society or conditions. The shapes of the four curves will be estimated independently using data from the appropriate society and condition, with four separate final probabilities allowed. Examples of possible developmental trajectories in this modelling framework are shown in Fig 2.

Each curve's shape is determined by two parameters, one of which is directly interpretable as the age at which individuals in that society are halfway between infancy and adulthood in terms of the norm acquisition, in the sense that their sharing rates are equal to the mean of the initial sharing rate and their society's adult sharing rates. The second parameter dictates how gradual or rapid the transition is. In mathematical terms, focusing on a single society and a single condition (i.e., intra- or inter-ethnic sharing), and denoting a participant's sharing choice by y, their age by x, the universal "infant" sharing probability by $p_{infant}$, and the (society-specific, condition-specific) adult sharing probability by $p_{adult}$, then the full model specification

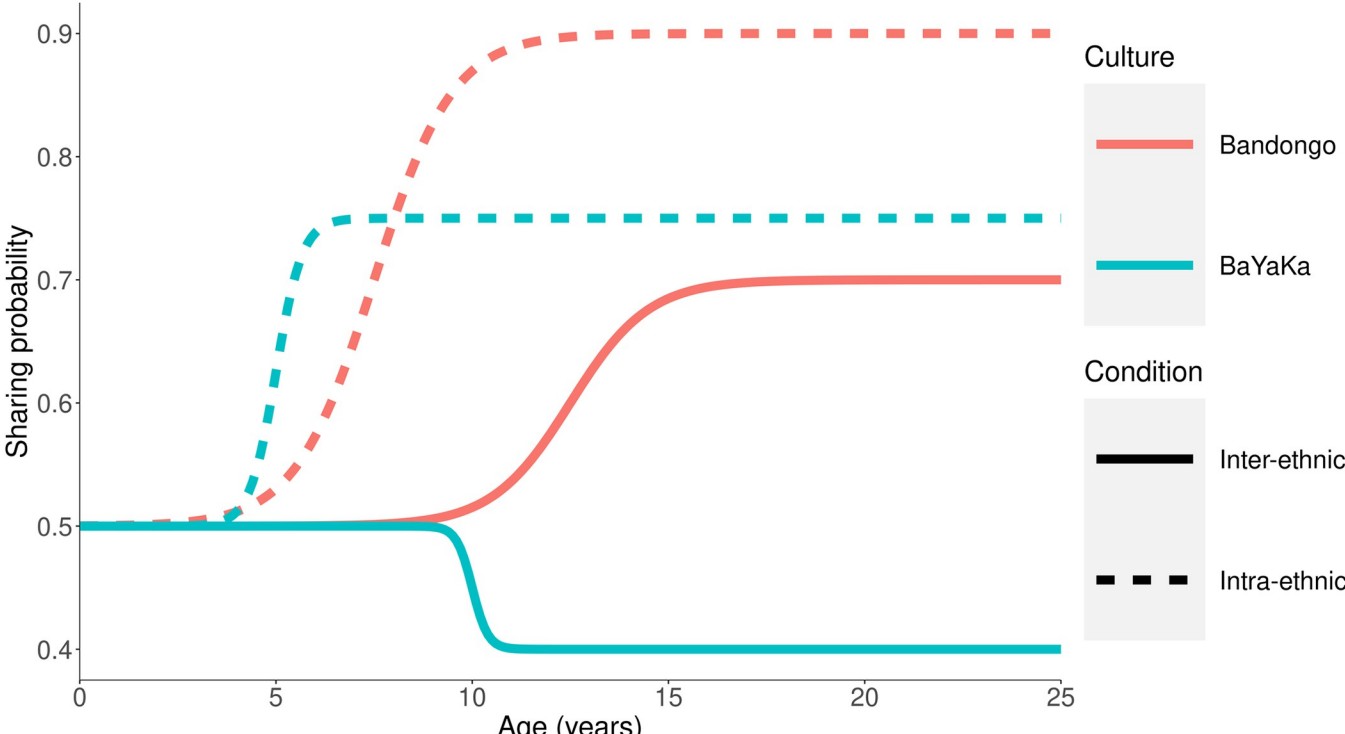

**Fig 2. Example developmental trajectories which can be expressed within the proposed modelling framework.** Colours correspond to societies and line types correspond to intra- and inter-ethnic sharing rates within each society. At age zero, all participants are expected to behave identically. By adulthood, each society is expected to have distinct intra- and inter-ethnic sharing norms. The change between these endpoints can happen at different ages and different rates across both society and condition. Here, BaYaKa children acquire norms suddenly (blue curves) while Bandongo norms shift more gradually (red curves).

including priors is:

$$p_{\text{infant}} \sim \text{Beta}(1.5, 1.5)$$
$$p_{\text{adult}} \sim \text{Beta}(0.75, 0.75),$$
$$\begin{bmatrix} a \\ b \end{bmatrix} \sim \mathcal{N}\left( \begin{bmatrix} 10 \\ 0.2 \end{bmatrix}, \begin{bmatrix} 15 & 0.75 \\ 0.75 & 0.2 \end{bmatrix} \right)$$
$$y \sim \text{Bernoulli}\left( p_{\text{infant}} + \frac{p_{\text{adult}} - p_{\text{infant}}}{1 + exp(-b(x - a))} \right)$$

For all four curves, a multivariate Normal prior distribution, truncated to allow only positive values, will be placed on the two parameters, with a positive correlation such that curves where development "starts later" are also more likely to be curves where development "happens quicker" to avoid development continuing into adulthood. Fig 3 shows 100 randomly sampled curves from our prior, showing the window from infancy to 20 years of age. A large majority of the curves have completed or very nearly completed development by the end of this window.

For any one of the four development curves, we can calculate a "completion age" corresponding to the age when 95% of the change between the infant and adult sharing probabilities has taken place. This has happened when $exp(-b_{ij}(x - a_{ij})) = 0.05$, i.e. at age $x \approx a + 3/b$. Fitting the model to data provides a joint posterior distribution for each of the a and b parameters, which translates to posterior distributions for the completion ages defined above and, most

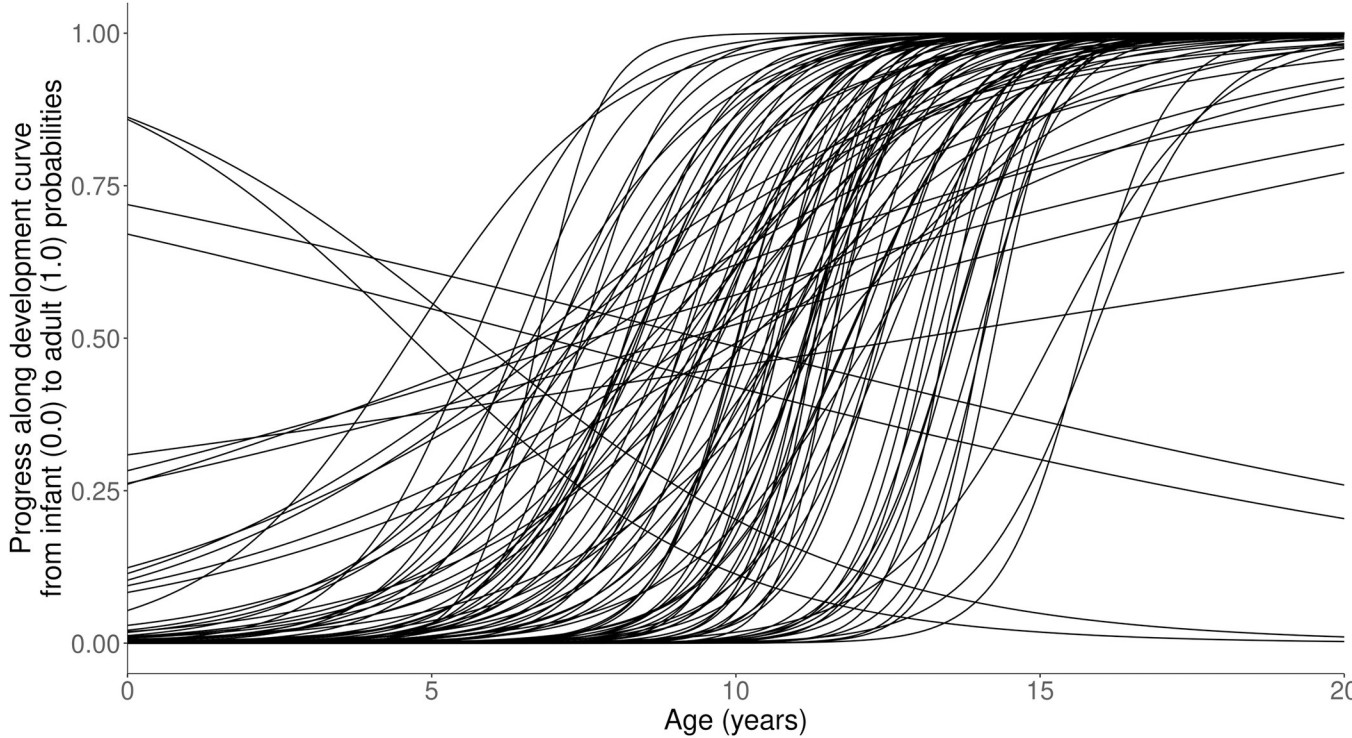

**Fig 3. Visualization of multivariate normal prior over developmental curve parameters, shown for the first 20 years.** The majority of curves approach complete transition to adult norms prior to the end of adolescence. Only 2 of the 100 randomly sampled curves do not show development being more than 80% complete by the age of 20.

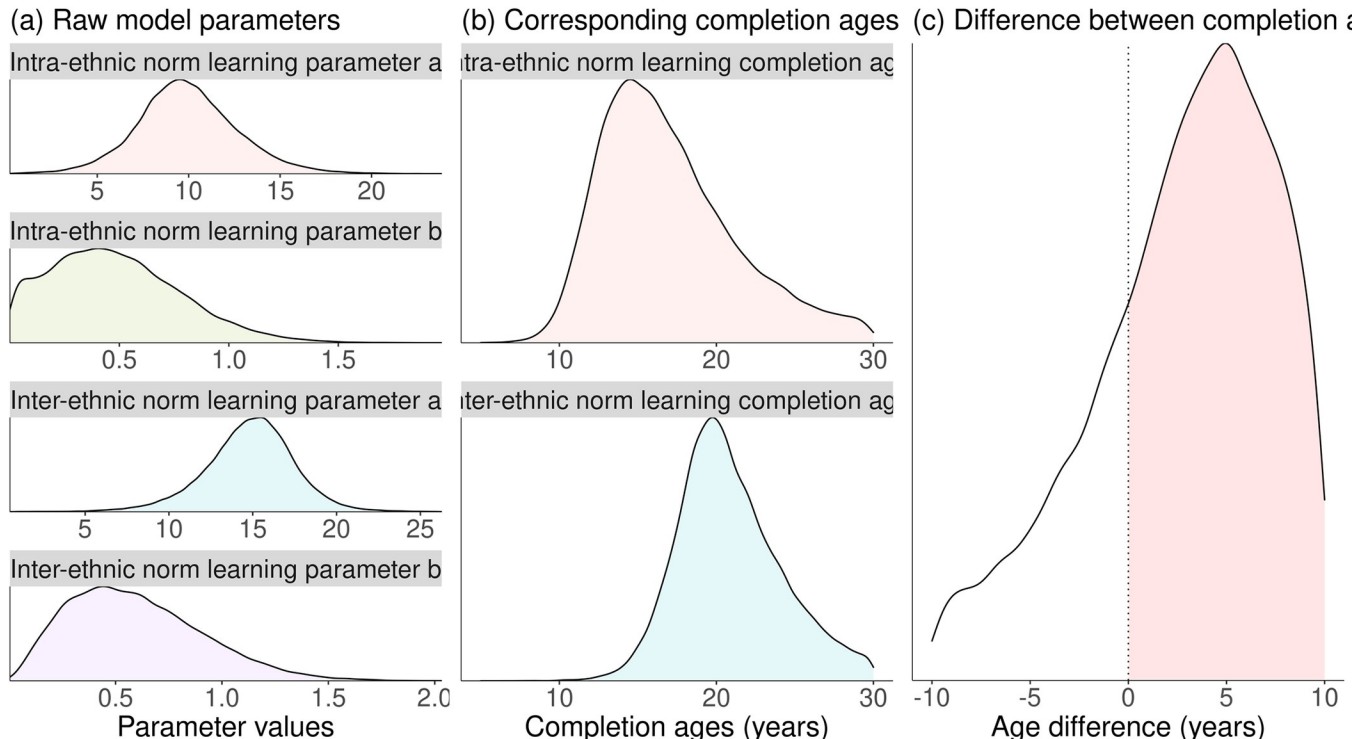

**Fig 4. Visualization of our interpretation of the fitted model for a single society.** The leftmost panel (a) shows the posterior distributions for the four separate model parameters, neither of which bares directly on our hypothesis. Each pair of posteriors for the a and b parameters determines a posterior distribution for the age at which acquisition of a particular norm is completed, as shown in the centre panel (b). This pair of distributions in turn defines a posterior distribution over the difference in completion ages. This distribution is shown in the rightmost panel (c), computed by subtracting the inter-ethnic completion age from the intra-ethnic completion age, such that positive values are consistent with our hypothesis that intra-ethnic norms are acquired earlier. The red-shaded area in panel (c) represents the posterior probability that our hypothesis is correct.

importantly for our purposes, the *difference* between these ages. The data's degree of support for our hypothesis that intra-ethnic sharing norms are acquired earlier than inter-ethnic sharing norms can be quantified as the posterior probability that the inter-ethnic completion age is older than the intra-ethnic completion age, i.e., the posterior probability that the former value minus the latter is greater than zero. Note that this value may be high even if the individual 95% HPD intervals for the two different completion ages overlap. Fig 4 shows a visualization of this progression for a single society from posterior distributions over individual model parameters to a single posterior probability value indicating support for our hypothesis. Values of this posterior probability close to 1.0 indicate strong support for our hypothesis, while values close to 0.0 indicate strong contradiction of the hypothesis and values close to 0.5 indicate that the data is inconclusive. We will report a posterior probability exceeding 0.75 as support for our hypothesis. Regardless of the posterior probability obtained, we will report posterior mean and 89% HPD intervals for the completion ages corresponding to all cultures and norms.

To assess the ability of a dataset of our proposed size to answer the question of which norm develops faster (i.e., power), we produced 30,000 simulated datasets containing sharing choices for 60 children and 40 adults from a society. The data were generated according to the statistical model shown above, with parameters values chosen such that intra-ethnic sharing norms were always acquired before inter-ethnic sharing norms (i.e., had a younger 95% completion age), with the time difference between acquiring the two norms varying between 0.5 and 6 years of age. Because accurately estimating the developmental trajectory is easier when the

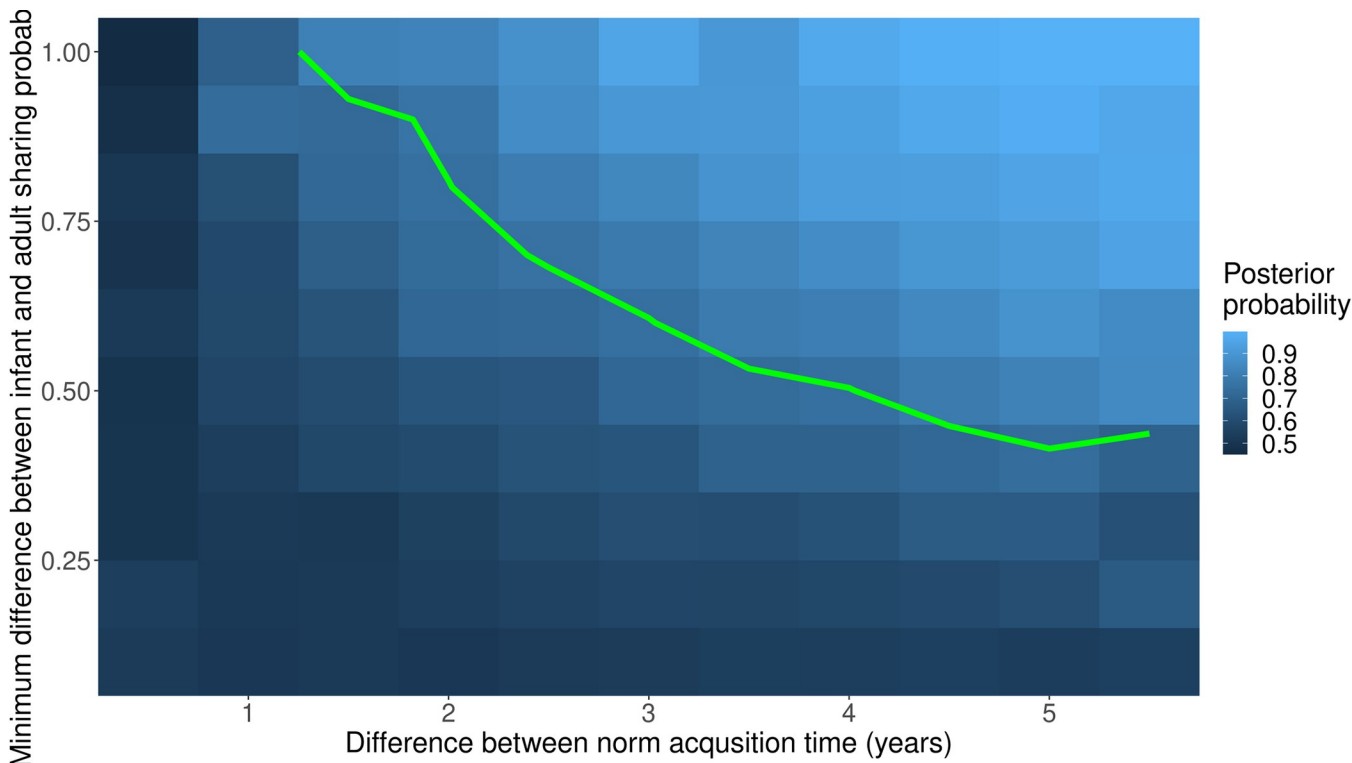

**Fig 5. Mean posterior probability that the developmental process for intra-ethnic norm acquisition reaches the 95% complete point earlier than the corresponding process for inter-ethnic norms, based on 30,000 simulations in total.** The red line shows our chosen threshold of 0.75. When the difference between norm acquisitions times (shown in years on the x-axis) is only 1 year or less, our sample size is not able to detect this with confidence. Greater differences can be detected if the adult norms differ sufficiently from the infant behaviour. The y-axis corresponds to the minimum absolute difference between the infant sharing probability and the two adult norm sharing probabilities.

difference between the initial infant sharing probability and the adult norms is greater, we also varied parameters to explore the impact of this on our ability to draw strong conclusions. Fig 5 shows the average posterior support for our hypothesis for all simulated datasets corresponding to a particular difference in acquisition age (x-axis) and a particular minimum separation between the initial sharing probability and both adult sharing probabilities (y-axis). The green contour line corresponds to mean posterior support of 0.75. The contour suggests that differences in acquisition completion times less than one year are unlikely to be reliably detected regardless of how distinct the sharing probabilities are, while differences greater than four years may be detected even if the developmental shifts in rates of sharing are relatively subtle. In the absence of relevant previous studies, it is difficult to assess in advance how likely our planned sample size is to yield a confident detection of difference in norm completion ages. Still, the above analysis shows such a detection is certainly not impossible in principle. Note that even if we are unable to confidently identify a difference in norm completion ages from these data, other differences in the developmental trajectory may still be identifiable. Further, the fitted model will provide estimates with quantified uncertainty of all aspects of the developmental process for both societies, including the various sharing rates. These estimates are still potentially ethnographically informative, and can be combined or compared with the results of ethnographic interviews. In particular, if the model is able to make confident statements about some aspects of norm acquisition which corroborate the accounts given in interviews, then those interviews may help to disambiguate other aspects where the model is less confident.

## Proposed timeline

We will train research assistants in July-August 2023. After a refresher in experimental methods, data will be collected in early 2024, when most villagers have returned from their fishing camps, and thus, the population of each village is at its highest. Data will be analyzed in March-April 2024. We aim to submit the final manuscript with data and analyses in September 2024. Our timeline will adjust to European and Congolese COVID-related regulations regarding safe travel.

## Supporting information

**S1 File. Inclusivity in global research.**
(DOCX)

## Acknowledgments

Thanks to Vidrige Kandza for comments.

## Author Contributions

**Conceptualization:** Sarah Pope-Caldwell, Sheina Lew-Levy, Adam H. Boyette, Kate Ellis-Davies, Harriet Over, Bailey R. House.

**Data curation:** Luke Maurits.

**Formal analysis:** Luke Maurits.

**Methodology:** Sarah Pope-Caldwell, Sheina Lew-Levy, Adam H. Boyette, Bailey R. House.

**Project administration:** Sarah Pope-Caldwell, Sheina Lew-Levy.

**Supervision:** Sheina Lew-Levy, Daniel Haun.

**Validation:** Luke Maurits.

**Visualization:** Sarah Pope-Caldwell, Luke Maurits.

**Writing – original draft:** Sarah Pope-Caldwell, Sheina Lew-Levy, Luke Maurits, Adam H. Boyette, Harriet Over, Bailey R. House.

**Writing – review & editing:** Sarah Pope-Caldwell, Sheina Lew-Levy, Daniel Haun.

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
