## [Decision Letter · Decision Letter 0]

29 Jun 2022

PONE-D-22-11734The social learning and development of intra- and inter-ethnic sharing norms in the Congo Basin: A registered report protocol PLOS ONE

Dear Dr. Lew-Levy,

Thank you for submitting your manuscript to PLOS ONE. I typically like to get at least two reviews before making a decision, but after protracted failed attempts at recruiting suitable reviewers feel that sufficient time has lapsed and rather than delay matters further will proceed based on the detailed feedback already provided.

As you can see, the reviewer is positively disposed to your proposed work, as am I. They set out clearly areas that are in need of consideration, and I would like to see you address each of these. I thus invite you to resubmit a revised manuscript. In the spirit of transparency, I intend sending any revision back to the reviewer for comment.

We look forward to receiving your revised manuscript.

Kind regards,

Mark Nielsen, Ph.D.

Academic Editor

PLOS ONE

Journal Requirements:

 "SLL was funded by a postdoctoral fellowship from the Alexander von Humboldt Foundation. DH was funded by the Max Planck Society for the Advancement of Science."

6. Please ensure that you include a title page within your main document. You should list all authors and all affiliations as per our author instructions and clearly indicate the corresponding author

Reviewers' comments:

Reviewer's Responses to Questions

**Comments to the Author**

1. Does the manuscript provide a valid rationale for the proposed study, with clearly identified and justified research questions?

Reviewer #1: Yes

2. Is the protocol technically sound and planned in a manner that will lead to a meaningful outcome and allow testing the stated hypotheses?

Reviewer #1: Yes

3. Is the methodology feasible and described in sufficient detail to allow the work to be replicable?

Reviewer #1: Yes

4. Have the authors described where all data underlying the findings will be made available when the study is complete?

Reviewer #1: Yes

5. Is the manuscript presented in an intelligible fashion and written in standard English?

Reviewer #1: Yes

6. Review Comments to the Author

You may also provide optional suggestions and comments to authors that they might find helpful in planning their study.

Reviewer #1: First, I'm really excited to see how this project turns out. It *needs* to be done, and this is the right team to do it. I have numerous compliments for this registered report:

- The introduction was excellent -- well-researched, thorough, and fun to read (I learned a lot about the study site!). I hope to see it again in the eventual publication.

- The team paid thorough attention to exclusion criteria and sample size. Their use of video cameras as backups is important.

- The method they're using for the Dictator Game was adopted from team member House's lead-authored paper (2013) -- the asocial condition from that paper worked well with a neighboring population (and many other populations, across ages, around the globe).

- Their comprehension checks are extremely clear and should be very straightforward for shy participants.

- This is a perfect use of logistic regression (have I ever seen better?) and the team made a clear commitment to a "significance" level of 0.7 (and while the rationale for that threshold is not given, I trust them to have picked a reasonable threshold given their expertise).

I have one larger suggestion/food-for-thought and some smaller ones:

- In this RR at least, the team appears to be giving far more brain time (if reflected by space used in the RR) to the DG relative to the interview. Given the team makeup, I'm sure they'll make good use of their opportunities in the field, but just in case this slipped under the radar, here's what I'm noticing: The team flagged the comparative dearth of ethnographic data on Bandongo social learning of sharing norms, but hasn't laid out any plans to remedy the situation. This seems important because the DG can only take us so far, telling us the timing of when a specific kind of sharing is acquired -- not trade, for example, and not multi-recipient sharing of parts of a carcass, but just division of a windfall between two people. While the team plans to learn how and from whom intragroup and intergroup sharing is learned, they're leaning hard on quantitative measures that work well with a Fisher's exact test. Ethnographic richness, including free responses from less-shy participants, could add ethnographic richness and help the researchers really understand the nature and life-course of norm acquisition. They could ask questions like:

-- In what context are norms acquired, for example? [Observation can happen in myriad places]

-- What does interethnic sharing mean to you? [...just to examine whether the researcher's interpretation of their observations of interethnic sharing matches what participants think are going on.]

-- Are kids learning differently today than they did in the past? And...

-- Are interethnic interactions between BaYaka and Bandongo *different* today than they were in the past? [Secular trends could be sliding past the researchers that would be informative to any differences between kids and adults.]

...of course this, and what I'm about to suggest below, could be difficult to pull off given the team is relying on research assistants, in which case these may be moot points.

- Relatedly, assuming child participants aren't *too* shy and just really want to get out of the research house after the DG wraps up, the researchers could think more about how to get broadly useful data from the post-game interview. For example, other yes/no questions the children could answer that could again add to ethnographic richness are things like:

-- Did you learn interethnic sharing from your mom?

-- Does interethnic sharing mean X to you?

...things that (in addition to their ethnographic observations) allow the team to check whether kids are learning in the same way as adults report *they* learned in the past in the interview -- again, thinking about secular trends and changes in intergroup interaction over time.

- This is a tall ask, and I don't expect the team to necessarily address it, but at least to think about it if they haven't already. The team is attending to things like participant shyness and mobility that might affect sample size. What about self-selection in this regard -- is there any reason to think that candidate participants who acquired interethnic norms earlier, through a different means, or (per my suggestions above) even have a different *set* of interethnic norms would be more mobile or less shy? This is what I would expect a priori, and I wonder if the researchers have any insight into whether self-selection on these lines (or others) would affect their results.

- Sample size will be sufficient if a few conditions hold: the transition from infant to adult state isn't very fast and/or the difference between intergroup and intragroup norm acquisition is four years. Four years is a while developmentally. Do the researchers have any thoughts on how to disambiguate between a quick transition vs similarity in timing of acquisition if results are inconclusive? For example, per the above, it seems that ethnography could really be an asset in that circumstance, including adult's interview responses about when they learned the relevant norms.

- Just to flag this: the team mentions ethnic markers at the outset and never addresses how markers come into play (or not) in BaYaka and Bandongo interactions, but implicitly leans on ethnic markers in the game design, treating red as one ethnic group and blue as another. Presumably markers are salient to BaYaka and Bandongo participants, as they seem to be (in some form or another) to all humans, but establishing this might check the ecological validity of the game (although I'm fine with the assumption that humans everywhere can work with this -- again, just flagging the assumption).

- At one point, the authors end a paragraph about intergroup interactions in children by saying "Inter-group cooperation may be enhanced in communities where strong social norms regulate inter-ethnic interactions" but don't provide citations. When the time comes, they can use two they already have -- Fearon & Laitin, and Bunce (although the 2018 Bunce paper may be better) -- and can also consider any paper about how religion sets norms for intergroup interactions (e.g., Islam in East Africa is a classic -- see Ensminger 1992 Making a Market) and some classic stuff by Barth (e.g., Barth 1956 Ecologic Relationships of Ethnic Groups in Swat, North Pakistan).

- Something to consider: the lag time between research assistant training and project commencement is a bit long (six months-ish). I get the sense that the team does this all the time, so I'm sure they know the best way to go about this, but a refresher for the RAs could be useful before things get started in early 2023.

All in all, again, a well-constructed team doing a project that very much needs to be done, with many laudable things in the RR. I look forward to reading and citing the end product!

7. PLOS authors have the option to publish the peer review history of their article (what does this mean?). If published, this will include your full peer review and any attached files.

Reviewer #1: No

---

## [Decision Letter · Decision Letter 1]

17 Oct 2022

The social learning and development of intra- and inter-ethnic sharing norms in the Congo Basin: A registered report protocol

PONE-D-22-11734R1

Dear Dr. Lew-Levy,

We’re pleased to inform you that your manuscript has been judged scientifically suitable for publication and will be formally accepted for publication once it meets all outstanding technical requirements.

Kind regards,

Mark Nielsen, Ph.D.

Academic Editor

PLOS ONE

Reviewers' comments:

Reviewer's Responses to Questions

**Comments to the Author**

1. Does the manuscript provide a valid rationale for the proposed study, with clearly identified and justified research questions?

Reviewer #1: Yes

2. Is the protocol technically sound and planned in a manner that will lead to a meaningful outcome and allow testing the stated hypotheses?

Reviewer #1: Yes

3. Is the methodology feasible and described in sufficient detail to allow the work to be replicable?

Reviewer #1: Yes

4. Have the authors described where all data underlying the findings will be made available when the study is complete?

Reviewer #1: Yes

5. Is the manuscript presented in an intelligible fashion and written in standard English?

Reviewer #1: Yes

6. Review Comments to the Author

You may also provide optional suggestions and comments to authors that they might find helpful in planning their study.

Reviewer #1: Bravo again to this team! I already thought this was a great registered report -- well thought-out and careful -- so I largely left food for thought for the researchers. They took the food-for-thought seriously and came back with an even more detailed, even more well-thought-out RR. I recommend acceptance -- and really look forward to seeing the products of the project.

7. PLOS authors have the option to publish the peer review history of their article (what does this mean?). If published, this will include your full peer review and any attached files.

Reviewer #1: **Yes: **Anne Pisor

---

## [Editor Report · Acceptance letter]

20 Oct 2022

PONE-D-22-11734R1 

The social learning and development of intra- and inter-ethnic sharing norms in the Congo Basin: A registered report protocol 

Dear Dr. Lew-Levy:

I'm pleased to inform you that your manuscript has been deemed suitable for publication in PLOS ONE. Congratulations! Your manuscript is now with our production department. 

Kind regards, 

on behalf of

Dr. Mark Nielsen 

Academic Editor

PLOS ONE